# Glioblastoma: A Retrospective Analysis of the Role of the Maximal Surgical Resection on Overall Survival and Progression Free Survival

**DOI:** 10.3390/biomedicines11030739

**Published:** 2023-03-01

**Authors:** Gabriele Polonara, Denis Aiudi, Alessio Iacoangeli, Alessio Raggi, Matteo Maria Ottaviani, Ruggero Antonini, Maurizio Iacoangeli, Mauro Dobran

**Affiliations:** 1Department of Neuroradiology, Università Politecnica delle Marche, Via Tronto 10/A, 60126 Ancona, Italy; 2Department of Neurosurgery, Università Politecnica delle Marche, Via Tronto 10/A, 60126 Ancona, Italy

**Keywords:** extent of surgical resection, FLAIR infiltration, glioblastoma, brain tumors, overall survival

## Abstract

Background: Glioblastoma (GBM) is the most common and aggressive primary brain tumor in adults; despite advances in the understanding of GBM pathogenesis, significant achievements in treating this disease are still lacking. The aim of this study was to evaluate the prognostic significance of the extent of surgical resection (EOR), beyond the neoplastic mass, on the overall survival (OS). Methods: A retrospective review of a single-institution glioblastoma patient database (January 2012–September 2021) was undertaken. The series is composed of 64 patients who underwent surgery at the University Department of Neurosurgery of Ancona; the series was divided into four groups based on the amount of tumor mass excision with the fluid-attenuated inversion recovery (FLAIR) abnormalities (SUPr-supratotal resection, GTR-gross total resection, STR-subtotal resection, BIOPSY). The hypothesis was that the maximal resection of FLAIR abnormalities may improve the overall survival compared to the resection of the visible T1 contrast-enhanced neoplastic area only. Results: In the univariate analysis, SUPr and GTR are correlated with the overall survival (*p* = 0.001); the percentage of total neoplastic removal threshold conditioning outcome was 90% (*p* = 0.027). These results were confirmed by the multivariate analysis. Conclusions: Maximal surgical resection, when feasible, involving areas of FLAIR abnormalities represents an advantageous approach for the OS in GBM patients.

## 1. Background

Glioblastoma (GBM) is the most common and aggressive primary brain tumor in adults, with an incidence of about 3.1 per 100,000 inhabitants/year increasing with age [1]. 

Glioblastoma standard treatment consists of a multimodal regimen that includes surgery as a first step to provide symptoms relief and histopathological and molecular information, followed by radio- and targeted chemotherapy [2].

Despite advances in understanding GBM pathogenesis and therapeutic regimen, after proper therapy, still today the median survival time is 13 months [3]. The main prognostic factors associated with a poor survival rate are a pre-operative low Karnofsky performance status score, age, brain location, the use of adjuvant radio-chemotherapy, and the presence of molecular factors such as the methylation promoter of the MGMT gene and the IDH-1 mutation [4]. Standard treatment consists of surgical resection, followed by radiotherapy and temozolomide chemotherapy. Temozolomide is given concurrently with radiotherapy and then for a further six months.

The most important clinical prognostic factors related to poor prognosis are: increasing age, poor performance status (PS), low degree of surgical resection of tumor. The extent of resection is one of the most important prognostic factors, and It’s related to an increasing overall survival and progression free survival. Patients with a major extent of resection have a better clinical outome, related to a better response to other therapy, such as chemoteraphy and radiotherapy [2,3].

Nowadays, the extent of surgical resection (EOR) has been identified as a prognostic factor for OS in GBM patients, but its quantification remains a matter of debate [2]. EOR is usually performed by post-operative contrast-enhancing MRI analysis, as highlighted by Lamborn et al. who classified the EOR into three categories: gross total resection (GTR), defined as tumor removal >90%; subtotal resection (STR) between 10–90%; and biopsy (B) <10% [5]. However, despite complete resection of the GBM contrast-enhancing portion, frequent relapses occur at the surgical not contrast-enhanced margins. These surgical cavity walls are considered potential sites of neoplastic infiltration and correspond to T2/FLAIR (fluid-attenuated inversion recovery) altered areas [6]. For this reason, the concept of supratotal resection (SUPr) was recently introduced, that is, resection of the contrast-enhanced tumor tending toward a maximal resection of the T2/FLAIR signal abnormalities [7].

In this study, we describe our single-center experience in terms of the impact of EOR on overall survival (OS) in GBM patients.

## 2. Materials and Methods

### 2.1. Patient Population and Neuroradiological Characteristics of the Tumor

This retrospective observational study involved 64 patients diagnosed with GBM who underwent surgery at our institution between 1 January 2012 and 1 September 2021. Patients’ data collection included clinical data and all MRI images on pre- and post-operative tumor localization, volume in contrast-enhanced T1 sequences, area volume of FLAIR abnormalities, and the presence of edema. MRI images were acquired through a 3 T scanner (Siemens) 48 h after surgery, and tumor volumes were analyzed via a semiautomatic region of interest (ROI) analysis with Iplan Cranial v3.0 software (Brainlab, Feldkirchen, Germany). The study included patients who had a single surgically removable neoplastic lesion GBM on pre-operative MRI while excluding those with multicentric GBM, other primary neoplastic disorders, metastases, and infectious diseases.

### 2.2. Extent of Resection and Outcome Evaluation

The primary end-point of this study was to evaluate patients’ OS in respect to tumor EOR defined as (pre-operative volume—post-operative volume/pre-operative volume) × 100% on post-operative contrast-enhancing MRI and classified into four categories (Figure 1). Three of them included those defined by Lamborn et al. such as GTR, STR, and biopsy, while the fourth was the SUPr, defined as a 100% resection of the contrast-enhanced tumor and 100% resection of the FLAIR signal abnormalities areas (Figure 2). In the case of tumor localization near eloquent brain areas, the EOR was modulated based on neurophysiological monitoring techniques, such as sensorimotor evoked potentials and electrocorticography. In these type of tumors, with localization close to eloquent areas, supramarginal resection was not possible. In these cases, we performed the biopsy or STR subtypes. The residual tumor volume (RTV) was defined as the total volume of altered areas still present in post-operative MRI images compared to the pre-operative imaging on contrast-enhanced (CE) T1-weighted spin-echo sequences. The OS was included in the prognostic evaluation along with the Karnofsky performance status (KPS) at one, three, six months, and then annually, and the evidence of adjuvant chemo-radiotherapy with temozolomide (TMZ) as for the Stupp regimen. The follow-up period was defined as the time interval between the date of the first surgery and patients’ death. Progression-free survival (PFS) was defined as the time interval between the first surgery and the radiological diagnosis of relapse, identified by the appearance of new enhancing areas at the surgical site or outside. Patients with relapse of tumor were not considered in this study.

### 2.3. Data Analysis

OS analysis was estimated using the Kaplan–Meier method. The univariate analysis was conducted using the log-rank test to examine the prognostic significance of each individual variable. The multivariate Cox proportional hazard regression stepwise method was employed to determine the independent relationship between a collection of variables and both overall and progression-free survival. The multivariate analysis was conducted to study the predictive factors of survival of the established endpoints. A *p* value < 0.05 was adopted as the threshold for statistical significance. The statistical analysis was carried out using the IBM SPSS package v.25.0.

## 3. Results

### 3.1. Study Population

Our sample was composed of 64 patients, including 34 men (53%) and 30 women (47%), aged between 25 and 84 years. The average age was 61 years, and the median was 62; 41 of them (64%) were aged less than or equal to 65 at the time of surgery against 23 patients over 65 (36%). All patients underwent surgery and 57 out of 64 patients underwent adjuvant chemo-radiotherapy. All patients treated with chemotherapy received administration of temozolamide. Of these, 56 underwent chemotherapy (87.5% of the total population), while radiotherapy was administered to 57 patients (89% of the total population). Of these, 1 patient underwent radiotherapy alone while the other 56 patients underwent concomitant therapy. Finally, 7 patients (11%) did not receive any adjuvant therapy (Table 1).

### 3.2. Neuroradiological Findings

All the patients in our group had a relapse, but those who underwent a second operation were 9 (14%). The SUPr resection group includes 14 patients (22%), the GTR group includes 10 (16%), the STR group includes 35 (55%), and the biopsy group includes 5 (8%). The median value of the pre-operative volume of the mass calculated in T1 was 40.1 cm³; the median of the FLAIR area volume was 50.3 cm³. In patients undergoing GTR, the median of the residual volume in T1 was 1.92 cm³ with a median of the volume in post-operative FLAIR of 26.2 cm³. In patients undergoing STR, the median of the residual volume in T1 was 10.5 cm³ and the median of the volume in post-operative FLAIR was 26.4 cm³. In the last group, namely, that of BIOPSY, the median of the residual volume in T1 was 17.3 cm³, and the median of the volume in FLAIR was 14.0 cm³. In the GTR group, the median of the volume resection percentages of the FLAIR area was 39%, and the median of the volume resection percentages of the T1 area was 94.0%. The median of the total resection rates in the STR group was 70%. The number of patients who received ≤90% total resection was 37 (58%) and >90% 27 (42%), see Table 2. Figure 1 and Figure 2 show the results of the measurement of the neoplastic lesion on MRI images.

### 3.3. Patient Outcomes

The long-surviving patients, i.e., those who had an OS over 60 months (5 years) were 2 (3%). The mean OS value of the SUPr group was 37.3 months with a median OS of 24 months, the GTR group had a mean OS value of 20.7 months and a median OS of 16 months, the STR patients had a mean OS value of 14.2 months and a median OS of 14 months, and, finally, the biopsy group presented an OS of 9.4 months and a median OS of 10 months. The mean PFS of patients undergoing SUPr was 12.6 months with a median value of 8 months; the GTR group had a mean PFS value of 10 months and a median PFS of 9.5; the STR group had a mean PFS of 5.9 months and a median value of 5 months; finally, the biopsy group had an average PFS of 3.2 months and a median PFS of 3 months (Table 3).

### 3.4. Statistical Results

In this study, patients with a total removal rate of ≤90% had a shorter overall survival than those with a percentage >90%, with statistical significance (*p* = 0.018), and these data are confirmed in the literature [7]. Finally, patients with surgical macroscopical tumor residue had a shorter overall survival with respect to those who did not [8,9,10]. The median values are, respectively, 14 vs. 18 months (*p* = 0.019). As documented in the literature, patients who were administered adjuvant chemotherapy had a higher median survival than those who did not [11]. In this study, median survival values were 17 vs. 4 months with *p*-value < 0.0001, respectively. Patients undergoing adjuvant radiotherapy had a higher median survival than those who did not. Median survival values were 17 vs. 3 months (*p*-value < 0.0001).

In the multivariate analysis with Cox regression, all values maintained statistical significance except the 1-month KPS. KPS > 70 at 1 month has a *p*-value of *p* = 0.074, at 3 months *p* = 0.011, and at 6 months *p* < 0.0001. Adjuvant therapy with chemotherapy and radiotherapy resulted in a *p* < 0.0001 in both cases. Resection >90% had a *p*-value of 0.027, the residual in the surgical site *p* = 0.028, the thickness of the capsule >15 mm *p* = 0.013. Finally, the statistical significance relating to the impact of the type of resection (SUP, GTR, STR, BIOPSY) in overall survival is very important with a *p*-value = 0.001 (Table 4).

## 4. Discussion

Still today in the literature, the role of tumor resection on OS in patients with GBM is debated, but recently, some studies document an increasing survival in patients undergoing a complete surgical resection of the contrast-enhancement area in T1 with a minimal residual tumor volume (RTV) compared to less extensive resections [2,12]. Some studies suggest a good removal rate between 78 and 89% of the contrast-enhancement area in T1 [8]. Sanai et al. documented that the median of survival increases was proportionate to the extension of the resection, with an OS of 12.8 months with EOR ≥ 80%, 13.8 months with EOR ≥ 90%, and 16 months with 100% EOR [13,14].

On the contrary, despite the complete resection of the entire area of contrast-enhancement in T1, it was observed that the relapse rate and the Disease Progression Free remains unchanged. 

In this light, it was proposed to consider a tumor removal that tends to the maximum resection of the area of T2/FLAIR abnormalities as well. This surgical strategy (SUPr) has been undertaken primarily for low-grade gliomas, while its use in GBM is still poorly investigated. Based on these observations, we investigated whether a SUPr-like resection for tumors in non-eloquent areas is associated with an advantage in overall survival compared to minor resections of the FLAIR area (GTR), to resections limited to the area of contrast enhancement in T1 (STR) or even in BIOPSY. It was studied how OS values change between the four-resection groups. The median of OS in SUPr patients is 24 months; in GTR patients, it is 16 months; in STRs, it is 14 months; and in the BIOPSY group, it is 10 months. The median of the OS of the whole sample is 16 months and the average is 20 months. The result is statistically significant both in univariate and in multivariate analysis.

Patients who received a SUPr-type extended resection had a higher median survival than all other classes. Median survival of SUPr patients was 24 months vs. 16 months of GTR vs. 14 months of STR vs. 10 months of BIOPSY. The test of equality of the survival distributions for the four groups reported a value of *p* = 0.003; therefore, the results were statistically significant. These data are supported by several studies that investigated and compared the impact on survival of SUPr-type resection with other types of less large surgical removals [9,14,15,16,17]. 

Several retrospective studies suggested that at least 78–89% of the contrast-enhancing tumor volume represents the resection target for a real survival advantage. The possible cause is the infiltration of GBM cells beyond the contrast-enhanced margins on MRI, as highlighted by biopsy specimens [18]. In patients with a tumor in an eloquent area where supramarginal resection cannot be achieved, the prognosis is poor. This is related to the impossibility of obtaining extensive excision in order to preserve neurological function.

As reported in many studies, the post-operative KPS is related with the outcome and our study confirmed this data. Patients with KPS ≤ 70 one month after surgery have a lower median survival than those with KPS > 70. The median OS is 11 vs. 16 months. However, the long rank test did not demonstrate statistical significance: *p* = 0.081. Patients with KPS ≤ 70 three months after surgery have a lower median survival than those with KPS > 70. The median OS is 7 vs. 17 months. In this case, the result is significant with *p* = 0.012. Patients with KPS ≤ 70 six months after surgery have a lower median survival than those with KPS > 70. The median OS is 11 months vs. 19 months. In addition, in this case, the result is statistically significant with *p* < 0.0001. This relationship may be explained with the possibility to complete the cycle of RT and chemotherapy only if the post-operative general conditions of the patient are good [5,19,20,21].

An association between the extent of resection and IDH mutation was not demonstrated due to the small sample of IDH-mutated patients. This could be explored in further studies with larger samples. 

The main limitations of this study are the retrospective nature of the investigation and the relatively low number of patients who underwent a SUPr-type resection. To overcome these limitations and make the study as rigorous as possible, only patients were selected with about the same timing of MRI investigations and patients operated upon with neuronavigation assistance.

## 5. Conclusions

This study documents how the resection of the entire tumor contrast-enhancement area has an advantage in overall disease survival compared to less extensive resections [22]. The tumor mass resection and the extension towards the MRI-FLAIR abnormalities region can represent a promising strategy capable of influencing the survival of people with glioblastoma. The study underlines the importance of a careful pre-operative evaluation of the neuroradiological parameters such as the MRI-FLAIR abnormalities region to obtain the maximum surgical resection for a better OS without the onset of additional neurological deficits. 

## Figures and Tables

**Figure 1 biomedicines-11-00739-f001:**
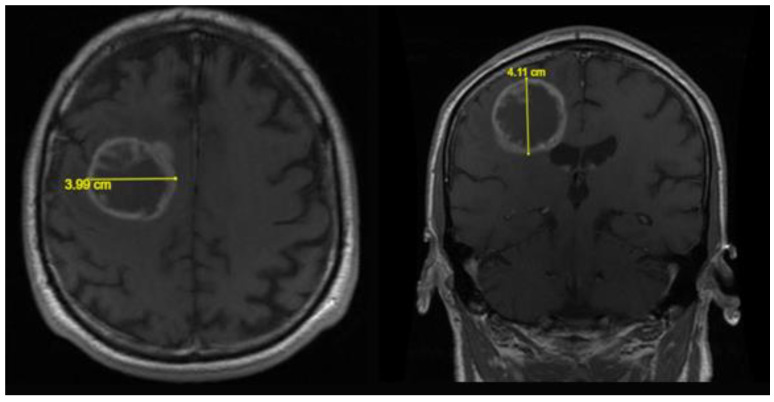
Axial and coronal T1 weighted MR images with measurement of the latero-lateral and cranio-caudal diameters of the tumoral lesion.

**Figure 2 biomedicines-11-00739-f002:**
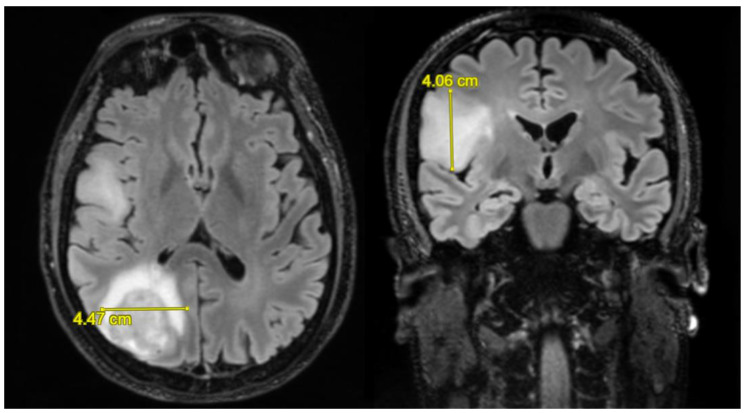
Axial and coronal FLAIR MR images with measurement of the latero-lateral and cranio-caudal diameters of the GBM.

**Table 1 biomedicines-11-00739-t001:** Study population.

	No. of Patients	%
Total	64	100
Gender		
Female	30	47
Male	34	53
Median Age	62	(Range 25–84 yrs)
Adjuvant treatment		
CHT	56	87.5
RT	57	89
CHT + RT	57	89
No adjuvant treatment	7	11
Long surviving (OS > 60 mos)	2	3

**Table 2 biomedicines-11-00739-t002:** Neuroradiological findings.

	No. of Patients	%
Surgical resection		
Supratotal resection (SUPr)	14	22
Gross total resection (GTR)	10	16
Subtotal resection (STR)	35	55
Biopsy (B)	5	8
Total resection		
>90%	27	42
≤90%	37	58
Tumor volume (TV)		
Median pre-operative T1 TV	40.1 cm³	(Range 1.2–201.8 cm³)
Median pre-operative FLAIR TV	50.3 cm³	(Range 10–183.1 cm³)

**Table 3 biomedicines-11-00739-t003:** Patient outcomes.

	Mean (Months)	Median (Months)
Overall survival (OS)		
SUPr group	37.3	24
GTR group	20.7	16
STR group	14.2	14
B group	9.4	10
1 month OS		
KPS ≤ 70		11
KPS > 70		16
3 months OS		
KPS ≤ 70		7
KPS > 70		17
6 months OS		
KPS ≤ 70		11
KPS > 70		19
Progression-free survival		
SUPr group	12.6	8
GTR group	10	9.5
Subtotal resection (STR)	5.9	5
Biopsy (B)	3.2	3

Supratotal resection (SUPr), gross total resection (GTR), subtotal resection (STR), biopsy (B), Karnofsky performance status (KPS).

**Table 4 biomedicines-11-00739-t004:** Data analysis.

Factors Analyzed	No. of Patients	Median OS Mos	Univ. (*p* Value)	Multiv. (*p* Value)
Total resection			0.018	0.027
>90%	27	18 ± 1.7		
≤90%	37	14 ± 2.8		
Surgical tumor residue			0.019	0.028
+	40	14 ± 2.2		
−	24	18 ± 1.1		
Extent of resection			0.003	0.001
SUPr	14	24 ± 4.8		
GTR	10	16 ± 2.4		
STR	35	14 ± 3.6		
BIOPSY	5	10 ± 2.2		
1 month KPS			0.081	0.074
KPS ≤ 70	11	11 ± 4.5		
KPS > 70	53	16 ± 1.2		
3 months KPS			0.012	0.011
KPS ≤ 70	16	7 ± 2.4		
KPS > 70	48	17 ± 3.7		
6 months KPS			<0.0001	<0.0001
KPS ≤ 70	23	11 ± 3.7		
KPS > 70	43	19 ± 1.2		
CHT			<0.0001	<0.0001
+	56	17 ± 1.3		
−	8	4 ± 1.4		
RT			<0.0001	<0.0001
+	57	17 ± 0.7		
−	7	3 ± 1.2		

## Data Availability

The data presented in this study are available on request from the corresponding author. The data are not publicly available due to privacy reasons.

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
