# Peer review of "Glioblastoma: A Retrospective Analysis of the Role of the Maximal Surgical Resection on Overall Survival and Progression Free Survival"

_biomedicines, 2023, doi:10.3390/biomedicines11030739_

Round 1

Reviewer 1 Report

1. To identify prognostic markers in glioblastomas, an immunohistochemical study is performed with the IDH1 antibody. It is likely that combining the results of immunohistochemistry with the volume of resection could give a more accurate prognosis. What do the authors think about this?

2. The volume of surgical intervention is affected by the anatomical location of the tumor itself and its surgical accessibility. In all cases included in your study, it was possible to increase the volume of resection?

3. None of the study participants were treated with temozolomide? Why? Combination chemoradiotherapy with temozolomide (daily temozolomide during radiation therapy followed by maintenance chemotherapy with temozolomide) is recommended for patients with glioblastoma as the first line of treatment after removal or biopsy (i.e., after histological verification of the tumor) (standard).

4. Did the volume of resection increase the number of functional disorders in patients? Will an increase in the volume of resection lead to an increase in survival, but a significant deterioration in the quality of life?

Author Response

1. To identify prognostic markers in glioblastomas, an immunohistochemical study is performed with the IDH1 antibody. It is likely that combining the results of immunohistochemistry with the volume of resection could give a more accurate prognosis. What do the authors think about this? It wasn’t demonstrated the association between extent of resection and IDH mutation due to the small sample of idh mutated patients. This will be be explored in further studies with larger samples.

2. The volume of surgical intervention is affected by the anatomical location of the tumor itself and its surgical accessibility. In all cases included in your study, it was possible to increase the volume of resection? In tumors with localization close to eloquent areas, supramarginal resection was not possible. In these cases we performed the biopsy or STR subtypes. 

3. None of the study participants were treated with temozolomide? Why? Combination chemoradiotherapy with temozolomide (daily temozolomide during radiation therapy followed by maintenance chemotherapy with temozolomide) is recommended for patients with glioblastoma as the first line of treatment after removal or biopsy (i.e., after histological verification of the tumor) (standard). All patients treated with chemotherapy received administration of temozolamide. In our study we didn't consider patients with different kind of chemioterapy. 

4. Did the volume of resection increase the number of functional disorders in patients? Will an increase in the volume of resection lead to an increase in survival, but a significant deterioration in the quality of life? In patients with a tumor in an eloquent area where supramarginal resection cannot be achieved, the prognosis is poor. This is related to the impossibility of obtaining extensive excision in order to preserve neurological functions.

Reviewer 2 Report

The paper is an interesting evaluation about the prognostic relevance of the extent of surgical resection beyond the neoplastic mass on the overall survival of patient affected by Glioblastoma.

The topic of the paper is worthy of investigation and well fits with the scope of the journal. Moreover, due to the serious impact of such tumour type on public health, the real implication of the study is of relevance. Thus, it is an opinion of this reviewer that the paper should be published, although some improvements are suggested to the Introduction and Materials and Methods sections as follows:

Introduction: major comment. The section should be improved by discussing the prognostic impact of different GMB treatment. This could help in better focusing the importance of the present study.

Materials and Methods: here, a minor suggestion is authors to consider the possibility to move Figure 1 and 2 to the results section.

Author Response

1) Introduction: major comment. The section should be improved by discussing the prognostic impact of different GMB treatment. This could help in better focusing the importance of the present study.

We can improve the introduction focusing the importance as predictor of survival of:

  • the extent of resection;
  • chemiotherapy and radiotherapy
  • radiological parameters

2) Materials and Methods: here, a minor suggestion is authors to consider the possibility to move Figure 1 and 2 to the results section.

It is possible to move the figures 1-2 in the result section, as suggested

Round 2

Reviewer 1 Report

The authors answered the reviewer's questions and made changes to the text of the article. I think that in its present form the article can be recommended for publication.

Author Response

Reviewer comment: The authors answered the reviewer's questions and made changes to the text of the article. I think that in its present form the article can be recommended for publication.

Answer: Thank you for the revision, We confirm that the manuscript has been modified as suggested.